# The evolution of same-sex sexual behaviour in mammals

José M. Gómez [1,2] ✉, A. Gónzalez-Megías [2,3] ✉ & M. Verdú [4] ✉

Same-sex sexual behaviour has attracted the attention of many scientists working in disparate areas, from sociology and psychology to behavioural and evolutionary biology. Since it does not contribute directly to reproduction, same-sex sexual behaviour is considered an evolutionary conundrum. Here, using phylogenetic analyses, we explore the evolution of same-sex sexual behaviour in mammals. According to currently available data, this behaviour is not randomly distributed across mammal lineages, but tends to be particularly prevalent in some clades, especially primates. Ancestral reconstruction suggests that same-sex sexual behaviour may have evolved multiple times, with its appearance being a recent phenomenon in most mammalian lineages. Our phylogenetically informed analyses testing for associations between same-sex sexual behaviour and other species characteristics suggest that it may play an adaptive role in maintaining social relationships and mitigating conflict.

Same-sex sexual behaviour, that is, any attempted sexual activity between members of the same sex[1–4], has been reported in over 1500 animal species, including all main groups from invertebrates such as insects, spiders, echinoderms, and nematodes, to vertebrates such as fish, amphibians, reptiles, birds, and mammals[1–3]. Same-sex sexual behaviour is particularly prevalent in nonhuman primates[5,6], where it has been observed in at least 51 species from lemurs to apes[7]. This sexual behaviour is not limited to one sex or to the existence of artificial conditions, as it has been observed in males and females both in captivity and in wild conditions[1–3,8]. Same-sex sexual behaviour is also frequent in humans, existing throughout most of our history and in many societies and cultures[9,10].

Same-sex sexual behaviour has been argued to incur higher costs than different-sex sexual behaviour[11]. First, sexual interactions with members of the same sex can have similar mating costs as sexual interactions with members of the opposite sex in terms of energy expenditure, time use, disease transmission, injuries, etc[8,12,13]. Second, because it does not directly contribute to reproduction, same-sex sexual behaviour additionally has the opportunity cost of not producing offspring, if same-sex sexual behaviour occurs instead of different-sex sexual behaviour[8,11]. For these reasons, the evolution and

prevalence of same-sex sexual behaviour is often considered a Darwinian paradox[3,4,11,14–16].

Several hypotheses have been proposed to explain the evolution and prevalence of same-sex sexual behaviour in human and non-human animals[2,8,11,17,18]. Some of these hypotheses are non-adaptive, suggesting that same-sex sexual behaviour is the consequence of mistaken identity[19,20], the limited availability of individuals of the opposite sex[21–23], the consequences of sexual frustration when individuals are refused by members of the other sex[20], or the by-product of selection acting on a separate trait, such as high sexual responsiveness[24]. A recently proposed hypothesis that is attracting much attention states that indiscriminate sexual behaviour (that is, the co-occurrence of different-sex sexual behaviour and same-sex sexual behaviour) is the ancestral condition for sexually reproducing animals and this explains the widespread occurrence of same-sex sexual behaviour in animals[3,16]. Under this view, indiscriminate sexual behaviour is proposed as the null hypothesis against which to test the occurrence of both different-sex sexual behaviour and same-sex sexual behaviour[3].

Contrasting with these non-adaptive explanations, other hypotheses are adaptive and suggest that same-sex sexual behaviour

¹Dpto de Ecología Funcional y Evolutiva, Estación Experimental de Zonas Áridas (EEZA-CSIC), Carretera de Sacramento s/n, La Cañada de San Urbano, 0-4120 Almería, Spain. ²Research Unit Modeling Nature (MNat), Facultad de Ciencias, Universidad de Granada, Granada, Spain. ³Dpto de Zoología, Facultad de Ciencias, Universidad de Granada, Avda Fuentenueva s/n, 18071 Granada, Spain. ⁴Centro de Investigaciones sobre Desertificación (CSIC-UV-GV), Crta Moncada-Náquera km 4.5, 46113 Moncada, Valencia, Spain. ✉e-mail: jmgreyes@eeza.csic.es; adelagm@ugr.es; miguel.verdu@ext.uv.es

can be directly favoured by natural selection[8,18]. For nonhuman mammals, two of the main adaptive hypotheses postulated to explain the origin, evolution and prevalence of same-sex sexual behaviour are:[18,25]

(i) Same-sex sexual behaviour contributes to establishing and maintaining positive social relationships[18]. According to this hypothesis, same-sex sexual interactions can serve to form and maintain bonds and alliances, and to facilitate reconciliation after conflicts between members of the same group[18]. This hypothesis predicts that same-sex sexual behaviour should be more frequent in social species than in non-social species[8].

(ii) Same-sex sexual behaviour contributes to diminishing intra-sexual aggression and conflict[8,18]. This hypothesis postulates that same-sex sexual interactions may serve to communicate social status and establish and reinforce dominance hierar-chies, thus preventing future conflicts, or may contribute to diverting aggressive behaviour toward courtship behaviour, providing subordinate males with greater opportunities to furtively copulate with females[18]. Because same-sex sexual behaviour is suggested to mitigate rather than completely eliminate aggressive behaviour, this second hypothesis predicts that same-sex sexual behaviour should be more frequent in species with aggressive and lethal intrasexual interactions than in more peaceful and nonlethal species. Lethal interactions are expressed in many species of mammals as the killing of conspecific adults (adulticide)[26]. This phenomenon appears to be mediated in males by mating competition and the establishment of dominance hierarchies. In females, on the other hand, the defence of resources and offspring mediates adulticide[26]. Therefore, due to these between-sex differences in motivation, the predicted asso-ciation between same-sex sexual behaviour and adulticide would be expected to occur mainly in males.

Most research to date has been focused on examining the adap-tive functions and disentangling the proximate causes of same-sex sexual behaviour within particular systems or species[4]. And several descriptive species-specific studies support these adaptive hypoth-eses. For example, same-sex sexual behaviour seems to facilitate reconciliation among group members in female bonobos (*Pan paniscus*)[27] and female Japanese macaques (*Macaca fuscata*)[28]. Simi-larly, same-sex sexual behaviour seems to serve to reinforce the alli-ance between small groups of male bottlenose dolphins (*Tursiops* spp.)[29], whereas it helps to strengthen dominance hierarchies in herds of American bison (*Bison bison*)[30]. Despite the value of these studies for inferring the reasons why same-sex sexual behaviour manifests in particular species, a deeper understanding of how this sexual beha-viour has evolved requires thorough testing of the adaptive hypoth-eses in a broader phylogenetic context[4,14–16]. Formal examination of these hypotheses needs exploration of the pattern of same-sex sexual behaviour across the phylogeny in order to infer the ancestral condi-tion and evolutionary history of same-sex sexual behaviour, and test-ing of their predictions using phylogenetically informed statistical analyses[3,14,15].

In this study, we examine the ability of the two non-exclusive adaptive hypotheses listed above to explain the evolution of same-sex sexual behaviour over all of Mammalia by using a phylogenetic approach. For this, we compile the existing information on mammalian same-sex sexual behaviour, defined as transient courtship or mating interactions between members of the same sex[2,18] (see Methods). Afterwards we infer its evolutionary distribution, reconstruct ancestral states and investigate whether the prevalence of same-sex sexual behaviour in mammals is influenced by the occurrence of sociality and/ or intraspecific lethal aggression.

## Results

### A preliminary cautionary note
We recognize that there may be some limitations in our database, and in our overall conclusions, caused by the lack of information on the sexual behaviour of many mammalian species and by the existence of incomplete data (false negatives). We have tried to overcome these caveats by controlling for the intensity of the research and conducting multiple statistical tests, although we are aware that this does not completely eliminate the limitations.

### Phylogenetic pattern of same-sex sexual behaviour
Same-sex sexual behaviour has been reported in 261 mammalian spe-cies (about 4% of the species) belonging to 62 families (about 50% of the families) and 12 orders (63% of the orders) (Supplementary Data 1). Same-sex sexual behaviour included courtship, mounting, genital contact, copulation and pair bonding[1,11]. In most cases, same-sex sexual behaviour was displayed as mounting and/or genital contact (87% of the species in our dataset), courtship (27% of the species), and pair bonding (24% of the species) (Supplementary Data 1). Same-sex sexual behaviour was mostly displayed by adults (same-sex sexual behaviour has been recorded in adults in 251 species and in young animals in 10 species; Fig S1, Supplementary Data 1)[25]. Likewise, 209 species dis-played same-sex sexual behaviour in wild or semiwild conditions (83% of the total sample), indicating that same-sex sexual behaviour is not a behaviour that emerges only in artificial conditions. In addition, whereas in some species same-sex sexual behaviour is incidental, occurring only under very specific situations, in about 40% of the species same-sex sexual behaviour is a moderate or even frequent activity during the mating season according to refs. 1,17 (Supplemen-tary Data 1).

Same-sex sexual behaviour appears to be equally frequent in both sexes in mammals, as female same-sex sexual behaviour has been recorded in 163 species and male same-sex sexual behaviour in 199 species. Nearly 52% of the species with same-sex sexual behaviour included in our dataset displayed both male and female same-sex sexual behaviour. To test for the presence of evolutionary correlation between male and female same-sex sexual behaviour, we used a recently updated mammalian phylogeny including 5747 extant and recently extinct mammals (see Methods). To control for the potential influence that among-species variation in research intensity may have in the outcomes of the phylogenetic correlation, we performed this analysis using four subsets of species: Subset I includes species where same-sex sexual behaviour has been recorded in any condition, either in captivity and/or in the wild; Subset II includes species where same-sex sexual behaviour has been recorded in the wild; Subset III includes species whose reproductive and sexual behaviour have been studied in the wild in more than one year or site; Subset IV includes species whose overall behaviour has been studied profusely (see Tables S1 & Figure S1 for the rationale and sizes of these four subsets of species and Methods for specific criteria used to differentiate among subsets). The results of the phylogenetic correlation analyses were consistent across subsets, and showed that male and female same-sex sexual behaviour are phylogenetically correlated across the tree ($\chi^2$ tests comparing cor-related and uncorrelated phylogenetic models ranged across the four subsets between $172.6 \pm 5.6$ and $476.9 \pm 3.0$ mean $\pm$ s.e.m, all $p$-values $< 0.0001$, Table S2). This finding indicates that male and female same-sex sexual behaviours are more likely to co-occur in the same species or clade than would be expected if these behaviours evolved independently.

To assess whether same-sex sexual behaviour is a behaviour dis-played just by one or a few distinctive groups of mammals, we explored its phylogenetic extent and calculated its phylogenetic signal (see Methods). We explored the phylogenetic signal of both male and female same-sex sexual behaviour using the D index for binary traits[31]. We checked for the potential effect of phylogenetic uncertainty by

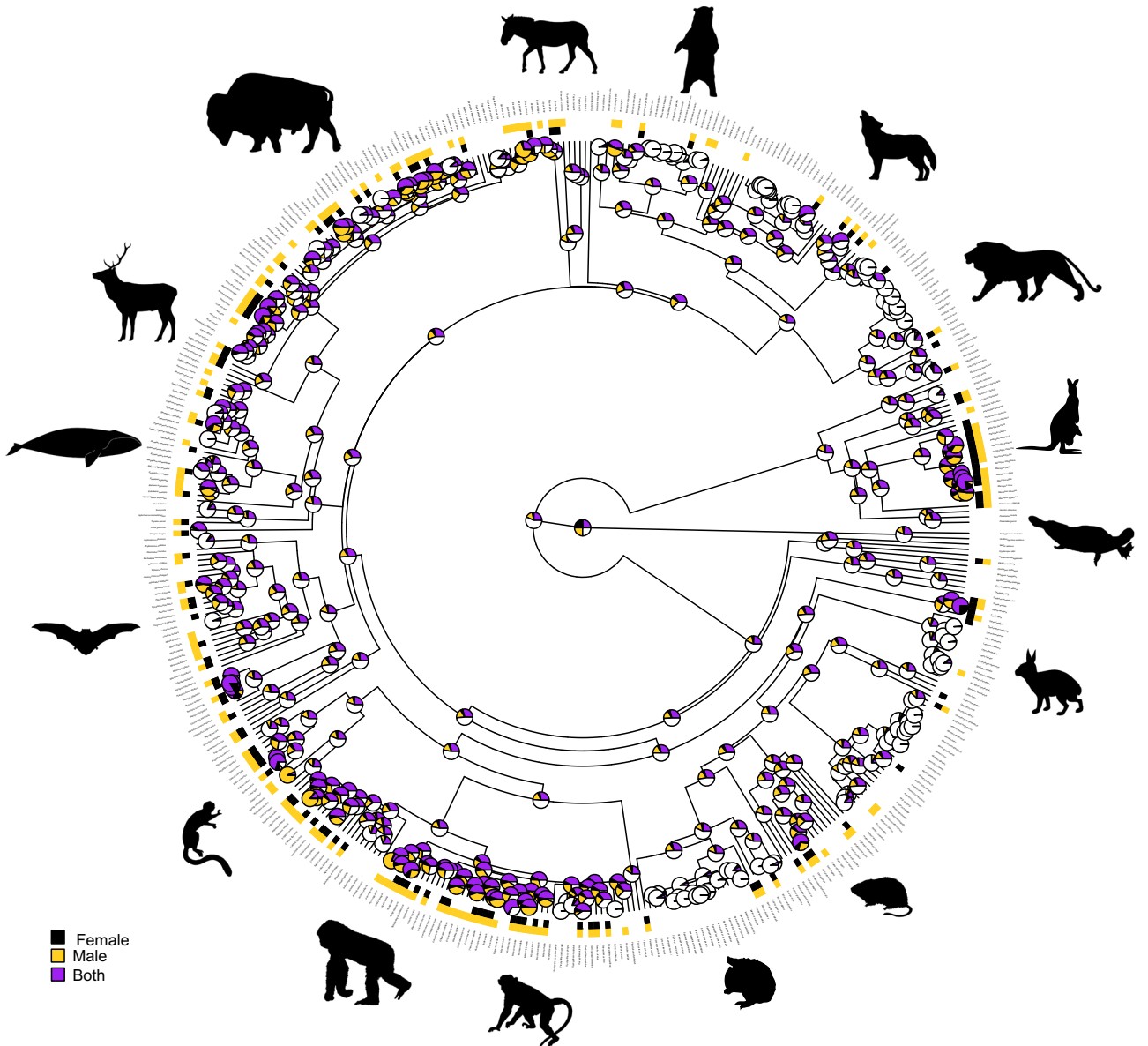

**Fig. 1 | Evolution of same-sex sexual behaviour in non-human mammals.** Phylogenetic distribution of the presence of same-sex sexual behaviour in males and females in the subset III (see methods). The state of the mammalian ancestral nodes was assessed using maximum likelihood estimation (black: same-sex sexual behaviour displayed by females; yellow: same-sex sexual behaviour displayed by males; purple: same-sex sexual behaviour displayed by both sexes). The silhouettes of representative mammals (downloaded from www.phylopic.org) illustrate the main mammalian clades. They have a Public Domain license without copyright (http://creativecommons.org/licenses/by/3.0).

Legend:
- ■ Female
- ■ Male
- ■ Both

repeating the analyses in 100 randomly chosen variations from the overall Mammalian tree (see Methods). In addition, we controlled for the potential effect caused by differences in research intensity by repeating the analysis in the four subsets described above. The results were consistent across all control methods (Table S3). We found a significant phylogenetic signal (i.e. $D < 1$) for both females ($D$ values ranging between 0.44 and 0.59, $p < 0.0001$) and males ($D$ values ranging between 0.63 and 0.83, $p < 0.001$ in all cases except for the subset IV) (Table S3). This outcome indicates that same-sex sexual behaviour is not randomly distributed across the mammalian phylogeny but tends to be frequent in some clades and rare in others (Fig. 1). Both male and female same-sex sexual behaviour was common in even-toed ungulates (Cetartiodactyla), carnivores, kangaroos and wallabies (Diprodontia), rodents and, above all, primates (Fig. 1). Nevertheless, D values were also significantly higher than expected under Brownian evolution ($D > 0$; Table S3), indicating that closely related species do not necessarily share this sexual behaviour.

## Is same-sex sexual behaviour an ancestral behaviour in mammals?

We reconstructed the ancestral condition of sexual behaviour and inferred the presence of same-sex sexual behaviour in the ancestral mammal and in the most recent common ancestors of the main mammal families in which same-sex sexual behaviour has been recorded in extant species. To control for the potential effect of phylogenetic uncertainty, we reconstructed the ancestral state of same-sex sexual behaviour using 100 randomly chosen trees (see above). In addition, to control for potential effects of dataset robustness, we reconstructed the ancestral state of same-sex sexual behaviour for each of the four subsets described above.

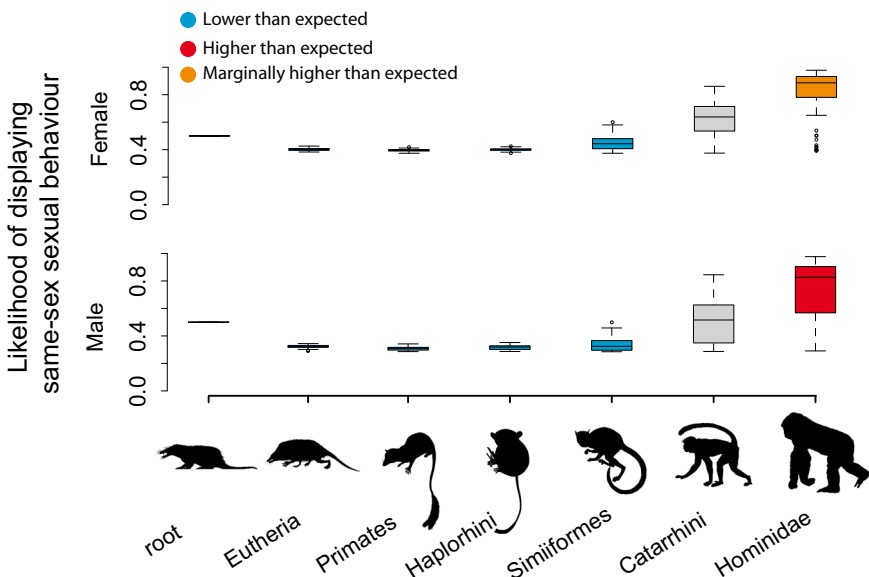

**Fig. 2 | Same-sex sexual behaviour from mammalian root to Hominidae.** Box-plots showing the likelihood estimation across 100 phylogenetic trees of the states of the most common recent ancestors (mrca) of the clades giving rise to the lineages of Hominidae (centre line, median; box limits, upper and lower quartiles; whiskers, minimum and maximum values). We used for the analysis the subset III. Red: significant probability ($p < 0.05$), according to a one-sided z-score test, of the ancestor displaying same-sex sexual behaviour; Orange: marginally significant probability of the ancestor of displaying same-sex sexual behaviour; Blue: significant probability of the ancestor of not displaying same-sex sexual behaviour; Grey: state of the ancestor was equivocal. For exact $p$-value see Table S8. The silhouettes illustrate the ancestors (downloaded from www.phylopic.org). They have a Public Domain license without copyright (http://creativecommons.org/licenses/by/3.0).

The results of the ancestral reconstruction analyses were consistent across all subsets (Table S4), indicating that the outcomes of these analyses are not significantly influenced by differences in research intensity. These analyses suggest that the likelihood that the most recent common ancestor of all mammals displayed either male or female same-sex sexual behaviour is equivocal (all likelihood values = 0.5 for whole Mammalian class, Table S4). That is, it is not possible to conclude with the existing information whether the ancestral mammal displayed same-sex sexual behaviour or not. However, the analysis suggests that the most recent common ancestor of all placental mammals did not display same-sex sexual behaviour, either among females (likelihood ranging between 0.08 and 0.32 across subsets, significantly lower than 0.5 in all cases) or among males (likelihood ranging between 0.10 and 0.40 across subsets, significantly lower than 0.5 in all cases according to a z-score test) (Table S4). This outcome (as well as those described below) may change if same-sex sexual behaviour is found in species in which, due to low sampling effort, it has not yet been detected.

We estimated the number of times same-sex sexual behaviour has been independently gained and lost during mammalian evolutionary history using 1000 iterations of stochastic character mapping[32]. The outcome of this analysis was consistent across the four data subsets and suggests that same-sex sexual behaviour has been gained and lost multiple times during Mammalian evolution with similar likelihood (Table S5).

We compared the average age of the internal nodes of the phylogeny where same-sex sexual behaviour was inferred as present with the average age of the nodes where it was inferred as absent. We expected some equivocal reconstruction as a consequence of the scarce information on same-sex sexual behaviour existing for many mammal species. To cope with this source of uncertainty, we generated a null distribution of likelihoods that was used as a baseline to decide the presence/absence of this behaviour for each ancestral node (see Methods for details), and repeated the analyses using the four subsets described above to control for research intensity. The results

of these analyses were again consistent across all subsets (Table S6). The ancestral nodes exhibiting male same-sex sexual behaviour were significantly younger than those not exhibiting male same-sex sexual behaviour. Thus, the average age of nodes with same-sex sexual behaviour ranged across subsets between $5.7 \pm 0.1$ and $7.6 \pm 0.1$ Myr. In contrast, the average age of nodes without same-sex sexual behaviour ranged across subsets between $6.8 \pm 0.1$ and $14.8 \pm 0.2$ Myr (all means significantly different at $p < 0.0001$ using a t-test; Table S6). A similar outcome was obtained for females, where the average age of nodes with same-sex sexual behaviour ranged across subsets between $5.6 \pm 0.3$ and $5.9 \pm 0.0$ Myr, whereas it ranged between $7.6 \pm 0.1$ and $13.9 \pm 2.5$ Myr for nodes without same-sex sexual behaviour ($p < 0.0001$ in all cases except for the subset I; Table S6). These results were consistent with our family-level analysis showing that both female and male same-sex sexual behaviour were probably absent in the ancestors of most families (Fig. S2, Table S7). As an example of this pattern, in Fig. 2 we illustrate how the probability of male and female same-sex sexual behaviour varies from the root of the mammals to the ancestor of Hominidae. It is readily observed that the probability of same-sex sexual behaviour remained low for most of evolutionary history, starting to increase at the origin of Old World monkeys (Catarrhini) and becoming significantly higher at the origin of apes (Table S8).

### Factors potentially facilitating the evolution of same-sex sexual behaviour

We explored whether the two focal adaptive hypotheses, the establishment and maintenance of social relationships or the diminishing of intrasexual aggression and conflict, can explain the occurrence of same-sex sexual behaviour in mammals. For this, we compiled information on sociality and intraspecific lethal aggression (estimated as adulticide committed by males or by females to any other individual) for the mammals included in our datasets[26,33,34]. Afterward, we tested whether these behavioural traits correlated with male and female same-sex sexual behaviour by means of comparative analyses[35,36].

**Table 1 | Outcome of the different analyses testing the effect of sociality and adulticide in male and female same-sex sexual behaviour and after controlling for sampling effort (see Methods and Tables S1 & S9 for full details)**

| | Sociality | Adulticide[a] | S x A |
|---|---|---|---|
| **Female same-sex sexual behaviour** | | | |
| Sampling effort as covariate (subset I) | 0.65** | −0.37 | 0.24 |
| Sampling effort as covariate (subset II) | 1.08*** | −0.15 | −0.24 |
| Sampling effort as weighting factor (subset I) | 1.89*** | 1.60** | −0.23 |
| Sampling effort as weighting factor (subset II) | 2.21*** | 1.57* | −0.28 |
| Sexual behaviour studied profusely (subset III) | 1.03*** | 0.62 | −0.42 |
| Overall behaviour studied profusely (subset IV) | 1.29*** | 0.41 | −0.43 |
| **Male same-sex sexual behaviour** | | | |
| Sampling effort as covariate (subset I) | 1.04*** | 0.64* | −1.32 |
| Sampling effort as covariate (subset II) | 1.04*** | 0.64* | −0.63 |
| Sampling effort as weighting factor (subset I) | 1.70*** | 2.53*** | −0.92 |
| Sampling effort as weighting factor (subset II) | 1.70*** | 2.63*** | −0.92 |
| Sexual behaviour studied profusely (subset III) | 1.17*** | 1.28*** | −1.28*** |
| Overall behaviour studied profusely (subset IV) | 0.10 | 0.99** | −0.04 |

[a]Female adulticide was included in those models testing the evolution of female same-sex sexual behaviour and male adulticide in those models testing the evolution of male same-sex sexual behaviour.

Adulticide by females was included in those models testing the evolution of female same-sex sexual behaviour and adulticide by males in those models testing the evolution of male same-sex sexual behaviour. We controlled in these analyses for the potential influence that research intensity may have in the relationship between same-sex sexual behaviour and the two explanatory variables using four complementary methods: (1) Including research effort as a covariate in the models[37–39]. This method was performed using both subsets I and II. (2) Including research effort as a weighting factor in the models[40,41]. This method was performed using both subsets I and II. (3) Running the models including those species whose reproductive and sexual behaviour have been studied profusely. This method was performed using subset III. (4) Running the models including those species where its overall behaviour has been studied profusely. This method was performed using subset IV (see Methods for details). The dataset sizes for each of these research intensity control methods and subsets of species are shown in Table S1.

The results were very consistent across all control methods (Table 1; Table S9), indicating that the relationships were robust to the effect of research intensity. We found that, after controlling for sampling effort and phylogeny, sociality was correlated with both male and female same-sex sexual behaviour in all cases (Table 1). Same-sex sexual behaviour was significantly more prevalent in social species than in non-social ones (Fig. 3).

The occurrence of adulticide was significantly correlated with same-sex sexual behaviour only for males (Table 1), male same-sex sexual behaviour being more prevalent in those species in which males are adulticidal irrespective of their sociality status (Fig. 3). In contrast, female adulticide did not correlate with female same-sex sexual behaviour in most of the analyses (Table 1; Table S9). In fact, the frequency of species displaying female same-sex sexual behaviour was similar between female adulticidal and non-adulticidal species both in social and non-social species (Fig. 3).

To discriminate whether the observed associations of same-sex sexual behaviour with sociality and adulticide reflect dependency relationships between these behaviours or independent evolutionary processes in the same direction, we used Pagel's directional test of trait evolution[42]. To do so, we tested which of the following four models of evolution best explains the empirical evidence: (1) A model assuming no dependency in the pairwise evolution of same-sex sexual behaviour with either of the other two behaviours. (2) A model assuming non-directional interdependency in these pairwise evolutions. (3) A model

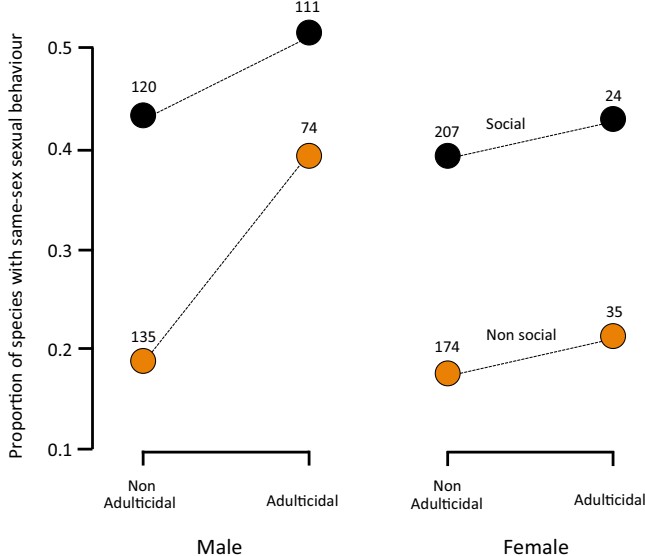

**Fig. 3 | Factors affecting evolution of same-sex sexual behaviour.** Proportion of species displaying same-sex sexual behaviour in social vs. non-social species and in adulticidal *vs.* non-adulticidal species. We used for the analysis the subset III. Sample size (total number of mammal species included in each group) is shown.

postulating that the evolution of adulticide or sociality was dependent on changes in same-sex sexual behaviour. (4) A model postulating that the evolution of same-sex sexual behaviour depended on changes in adulticide or sociality (see Methods for details).

This analysis suggested that the evolution of same-sex sexual behaviour and sociality were interdependent in both males and females, as the independent model did not obtain any support (Table 2). Moreover, according to the relative weights of the AICs, this analysis suggests that the evolution of same-sex sexual behaviour in both sexes depends on the presence of sociality (Table 2). In no case did the analysis support the possibility that the evolution of sociality depended on the presence of same-sex sexual behaviour (Table 2).

The directional test suggested that the relationship between the evolution of same-sex sexual behaviour and the evolution of adulticide differed between sexes. Male same-sex sexual behaviour and male

**Table 2 | Outcome of the analyses testing the evolutionary dependency relationships between same-sex sexual behaviour, sociality and adulticide (female adulticide in models testing the evolution of female same-sex sexual behaviour and male adulticide in models testing the evolution of male same-sex sexual behaviour)**

| Models of evolution | Likelihood | AIC | LRT[a] | AICw[b] |
|---|---|---|---|---|
| **Same-sex sexual behaviour versus sociality** | | | | |
| Males | | | | |
| Independence | −977.1 ± 1.5 | 1962.3 ± 3.0 | – | 0.00 ± 0.00 |
| Non-directional interdependence | −942.0 ± 1.4 | 1900.0 ± 2.8 | 70.3 **** | 0.40 ± 0.06 |
| Sociality depends on same-sex sexual behaviour | −963.3 ± 1.5 | 1938.6 ± 2.9 | 27.7 **** | 0.00 ± 0.00 |
| Same sex-sexual behaviour depends on sociality | −943.4 ± 1.4 | 1898.8 ± 2.8 | 67.5 **** | 0.60 ± 0.00 |
| Females | | | | |
| Independence | −895.6 ± 1.6 | 1799.1 ± 3.3 | – | 0.00 ± 0.00 |
| Interdependence | 861.6 ± 1.5 | 1739.1 ± 2.9 | 68.1 *** | 0.32 ± 0.03 |
| Sociality depends on same-sex sexual behaviour | 879.9 ± 1.7 | 1771.8 ± 3.5. | 31.4 *** | 0.00 ± 0.00 |
| Same-sex sexual behaviour depends on sociality | 862.6 ± 1.4 | 1737.3 ± 2.8 | 65.9 *** | 0.68 ± 0.03 |
| **Same-sex sexual behaviour versus adulticide** | | | | |
| Males | | | | |
| Independence | −1174.4 ± 1.6 | 2356.7 ± 3.1 | – | 0.00 ± 0.00 |
| Non-directional interdependence | −1128.6 ± 1.6 | 2273.3 ± 3.3 | 91.5 **** | 0.32 ± 0.03 |
| Adulticide depends on same-sex sexual behaviour | −1129.8 ± 1.6 | 2285.9 ± 3.0 | 89.1 **** | 0.01 ± 0.01 |
| Same-sex sexual behaviour depends on adulticide | −1136.9 ± 1.5 | 2272.3 ± 3.3 | 74.9 **** | 0.66 ± 0.03 |
| Females | | | | |
| Independence | −806.1 ± 1.4 | 1620.1 ± 2.7 | – | 0.06 ± 0.02 |
| Non-directional interdependence | −798.3 ± 1.4 | 1612.6 ± 2.8 | 15.6 | 0.47 ± 0.06 |
| Adulticide depends on same-sex sexual behaviour | −801.5 ± 1.4 | 1615.1 ± 2.9 | 8.3 | 0.20 ± 0.04 |
| Same-sex sexual behaviour depends on adulticide | −801.9 ± 1.5 | 1615.9 ± 3.0 | 9.3 | 0.27 ± 0.06 |

We used for this analysis the subset III.
[a]LRT comparing independence model against any of the other three models.
[b]Akaike relative weights of each model.

adulticide evolved interdependently, as all three models were statistically different from the independent model (Table 2). Moreover, according to the relative weights of the AICs, this analysis also suggests that this interdependency was directional, with the evolution of same-sex sexual behaviour depending on the presence of male adulticide (Table 2). No support was found for the existence of dependence of the evolution of male adulticide on changes in same-sex sexual behaviour (Table 2). For females, there was no statistical difference between any of the three dependent models and the independent model (Table 2), suggesting that the evolution of female same-sex sexual behaviour and the evolution of female adulticide were decoupled.

## Discussion

Same-sex sexual behaviour seems to be a common behaviour in mammals, recorded in about 5% of the species and 50% of the families, a frequency that appears to be higher than in other animal groups such as birds or insects[1,43–45]. This figure is probably underestimating the actual prevalence of same-sex sexual behaviour in mammals, since this behaviour has attracted the attention of behavioural ecologists and evolutionary biologists only recently[1–4,10] and it is probably underreported[11]. In fact, of the 22 mammalian species that are the subject of continuing long-term, individual-based field studies[46], same-sex sexual behaviour has been found in more than 80%[1,7,25]. All of this makes evident the necessity of increasing the number of studies on this overlooked behaviour and the convenience of controlling for several sources of uncertainty in order to reliably determine the prevalence of same-sex sexual behaviour in mammals.

It has been recently hypothesised that indiscriminate sexual behaviour, with the presence of same-sex sexual behaviour coexisting with different-sex sexual behaviour, is an ancestral condition for sexually reproducing animals[3]. Our ancestral reconstruction analysis

contrasts with this view for mammals and suggests that same-sex sexual behaviour is not an ancestral trait in this group of vertebrates, and may have evolved multiple times in several disparate lineages (although our study cannot conclude anything about ancestrality in other groups of animals). Furthermore, this analysis also indicates that those ancestral nodes exhibiting same-sex sexual behaviour are significantly younger than those ancestral nodes where this behaviour was absent. This finding agrees with some anecdotal observations. For example, despite appearing in some extant species, same-sex sexual behaviour was absent in the ancestors of Cebidae, Atelidae or Hylobatidae, three mammal families that seem to have originated very recently[47]. We fully recognize that these results may change in the future if same-sex sexual behaviour is studied more intensively and comes to be detected in many more species[6,25]. This again emphasises the need to study this sexual behaviour more profusely in mammals. However, it is difficult to predict the number and phylogenetic position of species in which same-sex sexual behaviour exists but has not yet been detected. For this reason, and with the data available to us, it appears that same-sex sexual behaviour has originated independently in many mammalian lineages.

These repeated evolutionary transitions to the same character state are an indication of convergent evolution[48]. Although convergence may occur from random evolution, convergence that is associated with similar selective environments is considered strong evidence of adaptive evolution caused by the operation of natural selection[49,50]. Finding these selective pressures may help to discern whether same-sex sexual behaviour is adaptive[51] and, in particular, to understand why this behaviour has evolved multiple times in mammals.

We found that the prevalence of same-sex sexual behaviour in mammals is associated with sociality. And the directional test of trait

evolution suggests that this covariance probably occurs because the evolution of same-sex sexual behaviour in both males and females has been contingent on shifts from solitary living to sociality. Albeit acknowledging that these findings may change if more data on same-sex sexual behaviour are reported, these results support the hypothesis that same-sex sexual behaviour has been favoured evolutionarily as a way to establish, maintain and strengthen social relationships that may increase bonds and alliance between members of the same group[8,18]. Furthermore, these results also suggest that same-sex sexual behaviour may have evolved also to facilitate post-conflict reconciliation, irrespective of its role of preventing intrasexual conflicts[18]. To be clear, our study suggests that one of the forces facilitating the evolution of same-sex sexual behaviour may be related to social bonds, but our study (like any other comparative or experimental study) does not conclude that this is the sole cause of the evolution of same-sex sexual behaviour.

The prevalence of same-sex sexual behaviour was also associated with adulticide, but only for males. The directional test of trait evolution indicates that evolution of same-sex sexual behaviour depended in males on the evolution of male adulticide. The difference between males and females in the relationship of same-sex sexual behaviour with adulticide supports the hypothesis that same-sex sexual behaviour has also evolved to mitigate intrasexual aggression and conflicts[18]. This is so because adulticide in mammals seems to be the consequence of intrasexual conflicts only in males, whereas it seems to be displayed by females primarily to protect their progeny against infanticidal conspecifics[26]. Consequently, it would be expected that, if same-sex sexual behaviour is a strategy directed to tempering intrasexual conflicts, this sexual behaviour should be related with adulticide only in males. Because the association was more intense in males than in females, we presume that adulticide was a stronger force triggering the evolution of same-sex sexual behaviour in males. If this hypothesis is confirmed, it seems that same-sex sexual behaviour mitigates rather than negates adulticide, as there are still many species that commit this type of aggressive behaviour[26].

Many descriptive studies on individual mammal species support our phylogenetic conclusions[18,25,27,52]. For example, same-sex sexual behaviour appears to be more common in social nonhuman primates forming multi-male/multi-female groups than in monogamous and polygynous species[7]. Likewise, same-sex sexual behaviour seems to facilitate reconciliation among group members in female bonobos (*Pan paniscus*)[27] and female Japanese macaques (*Macaca fuscata*)[28], to strengthen alliance between small groups of male bottlenose dolphins (*Tursiops* spp.)[29], and to help to reinforce dominance hierarchies in herds of American bison (*Bison bison*)[30]. In addition, it seems that the contingent evolution of same-sex sexual behaviour on sociality observed in mammals may also occur in other animal groups, such as male birds[44]. However, other studies have not found any evidence supporting these adaptive explanations. Same-sex sexual behaviour seems to be caused by mistaken identity in feral cats *Felis cattus*[20] or as a side effect of excitement in some primate and deer species[2,6,18,23]. Consequently, we cannot rule out the existence of other factors that may have also contributed to the evolution of same-sex sexual behaviour in certain mammalian lineages. Further studies framed within a phylogenetic context would be necessary to unravel the relative importance of each of these factors.

Contrasting with most other mammal families, same-sex sexual behaviour seems to have been present in the ancestor of Hominidae, an idea that has been suggested before[6]. This suggests that the evolutionary origin of same-sex sexual behaviour in humans can be traced back to the ancestor that we share with the other ape species. According to the existing evidence, the ancestor of Hominidae seems to have been a social species[33,53,54] exhibiting mostly male adulticide[26,34], two features that could have been facilitating the emergence of same-sex sexual behaviour at this time in our history.

However, same-sex sexual behaviour is operationally defined here as any temporary sexual contact between members of the same sex[2]. This behaviour should be distinguished from homosexuality as a more permanent same sex preference, as found in humans. For this reason, our findings cannot be used to infer the evolution of sexual orientation, identity, and preference or the prevalence of homosexuality as categories of sexual beings[2,11,18,45]. Nevertheless, even taking into account this cautionary note, by using phylogenetic inference, our study may provide a potential explanation on the evolutionary history of the occurrence of same-sex sexual behaviour in humans.

Putting all this evidence together, we envision the following evolutionary pattern of same-sex sexual behaviour in mammals. Sociality has evolved repeatedly in mammals from an ancestral solitary behaviour[33,46,53,54]. In mammals, social evolution is associated with the evolution of adulticide, mostly in males[26]. Due to the multiple benefits of sociality, many behavioural strategies have evolved to ensure the cohesion and stability of social groups[54]. Same-sex sexual behaviour could be one of these strategies[8]. Because individuals that engage in same-sex sexual behaviour can also practice different-sex sexual behaviour[1–4,11,18], they may improve their individual fitness through enhancing social relationships and mitigating conflicts by same-sex sexual behaviour to later have options to reproduce. We recognize that this scenario could be partially modified as more information on same-sex sexual behaviour in mammals is gathered. In addition, it does not preclude the contribution of other proximate mechanisms, like genetic mechanisms, practice, mistaken identity or excitement, causing the display of same-sex sexual behaviour in some species[18,20,23]. Briefly, our findings are consistent with the idea that, rather than a maladaptive[11,18] or aberrant behaviour[1,3], same-sex sexual behaviour in mammals is a convergent adaptation facilitating the maintenance of social relationships and the diminishing of intrasexual conflicts.

## Methods
### Definition of same-sex sexual behaviour
We define same-sex sexual behaviour from an operational point of view as any transient mating or courtship interaction between members of the same sex[2,18]. More specifically, same-sex sexual behaviour is any behaviour that is usually performed at some stage during reproduction with a member of the opposite sex, but which is instead aimed towards members of the same sex. These behaviours can include courtship, mounting, genital contact, copulation, pair bonding and the raising of offspring together[1,11]. Same-sex sexual behaviour as it is used here does not denote sexual orientation (i.e. an overall pattern of sexual attraction/arousal over time), sexual orientation identity (i.e. the sexual orientation that individuals perceive themselves to have), categories of sexual beings (i. e. homosexuals, heterosexuals, etc.), nor sexual preference[2,45].

### The database
We included the information on same-sex sexual behaviour appearing in refs. 1,2,7,25 (see Supplementary Data 1). We complemented this database by conducting computer searches including the terms (alone or in combination, and in British and American spelling) "mammal", "same-sex sexual behaviour", and "homosexual behaviour". We show in Fig. S1 a PRISMA flow diagram showing our systematic literature survey.

Sexual behaviour is a continuous rather than a dichotomous variable, with organisms and species engaging and participating in varying proportions of same-sex sexual behaviour[55]. However, because information on frequency and intensity of same-sex sexual behaviour is very scant, we have categorised in this study mammal species as those where same-sex sexual behaviour has been recorded or not.

We classified the social systems of mammalian species using the information appearing in refs. 26,33,34,54. The mammals were classified as (1) solitary, when breeding females forage independently in

individual home ranges and encounter males only during mating; (2) socially monogamous, when a single breeding female and a single breeding male share a common range or territory and associate with each other for more than one breeding season, with or without non-breeding offspring; or (3) group living, when several breeding females share a common range and forage or sleep together. Group-living species (which typically have polygynous or polygynandrous mating systems) include those where groups of breeding females are unstable, as in the case of ungulate herds or the roosting groups of some bats, as well as species where several breeding females associate with each other in stable groups for more than one breeding season, whether or not they always forage together[33]. We considered as social the species living in groups, and as non-social the species belonging to the other two categories. We obtained information for 2546 species[26,33,34].

We classified mammalian species according to lethal aggression as (1) adulticidal, when the killing, deliberately or incidentally, of a conspecific of any sex that has reached sexual maturity has been recorded; or (2) non-adulticidal, when this behaviour has not been reported. We used the information appearing in refs. [26,34].

Information on body size was obtained from panTheria[56], a dataset including information on 2909 mammalian species, Amniote[57], a dataset including information on 1548 mammalian species, EltonTraits 1.0[58], a dataset including information on 5731 extinct and extant species, and from Animal Diversity Web −https://animaldiversity.org/. In most cases body size was estimated as body mass (in grams), but in some species, we used alternative proxies of size to find dimorphism (body length, length of some body parts, centroid size from geometric morphometric analyses, etc.; see Supplementary Data 2).

### Controlling for the potential effect of research intensity on data analysis

To control for the potential influence of research intensity (i.e. how much a species behaviour has been studied scientifically) on our results, we performed all the subsequent phylogenetic and comparative analyses using four different proper subsets of species with the following increasing restrictive criteria:

**Subset I.** The entire set of species included in our original dataset. The species included in this set were all of those for which we were able to find data on the presence or absence of same-sex sexual behaviour irrespective of if this behaviour was observed in the natural conditions or in artificial conditions, such as laboratory, captivity, etc. This data set included 251 same-sex sexual behaviour species and 1470 species without recorded same-sex sexual behaviour (Table S1).

**Subset II.** This subset of species was obtained by removing all the species in which same-sex sexual behaviour has been observed only in artificial conditions (these species are marked as 'Captivity' in the column entitled 'Observed' in Supplementary Data 1) and keeping those species where this behaviour has been observed in wild conditions. This data set included 209 same-sex sexual behaviour species and 1470 species without recorded same-sex sexual behaviour (Table S1).

**Subset III.** For this subset, we included only those species whose reproductive and sexual behaviour has been studied profusely. This method was intended to minimise the effect of research intensity in the likelihood of observing same-sex sexual behaviour and getting false negatives (species scored as not having same-sex sexual behaviour despite actually displaying this behaviour occasionally). We retained from the previous dataset those species whose reproductive behaviour and sociality have been studied for several years or at several sites. To obtain this information, we conducted an additional computer search including the terms "sexual behaviour", "reproductive behaviour", "reproduction", and "fitness" for each of those mammal species. This data set included 205 same-sex sexual behaviour

species and 252 species without recorded same-sex sexual behaviour (Table S1).

**Subset IV.** For this dataset, we removed those species whose overall behaviour has been studied occasionally. Like the previous approach, this method intended to minimise the likelihood of getting false negatives. In this case, we retained from our original dataset those species with less than 1000 citations. By using this high number of citations as a cut-off threshold, this pruned dataset was composed exclusively of those species that have been studied intensely. Research intensity was estimated for each species as sampling effort, the number of citations per species[37,38]. For this, we collected data on citation counts for each species in our data set[37–39] by extracting the total number of references published on each species as reported in Google Scholar, using the species' scientific name (or their synonyms) and 'behaviour' as search parameters (data last accessed 10 July 2022). We used Google Scholar as the search engine because, having the same coverage as other databases for journal articles[59] and finding almost 90% (approximately 100 million) of all scholarly documents on the Web written in English[60], it also includes technical reports, books and conference presentations, has good coverage of non-English sources and Open Access articles, and is interdisciplinary, searching many topics at once. We included behaviour as a search parameter to ignore those studies made on non-behavioural topics, such as taxonomy, phylogeny, community or population ecology, ecological interactions, etc., that could bias our estimate of the research intensity made on the questions raised by this study. This data set included 154 same-sex sexual behaviour species and 238 species without recorded same-sex sexual behaviour (Table S1).

### Mammal phylogeny

The phylogenetic relationship among the mammals included in the database was built using the Faurby and Svenning phylogenetic tree that contains 5747 extant and extinct mammals[61]. We checked for potential effect of phylogenetic uncertainty by repeating all comparative analyses in 100 randomly chosen variations ("randomly chosen trees" hereafter) drawn from the Bayesian posterior distribution underlying the overall Faurby and Svenning tree (except for the phylogenetic correlation analysis, that was performed with 30 trees because of its large computational cost). In each phylogeny we pruned all species not included in the database and, in the few cases where a species was missing in the supertree (5 species), we selected the closest relative.

### Phylogenetic signal

Phylogenetic signals of male and female same-sex sexual behaviour were calculated with the phylo.D algorithm in the R package "caper"[62]. This algorithm compares the sum of changes in estimated nodal values of a binary trait (like same-sex sexual behaviour) along branches of the phylogeny against that expected for a random phylogenetic pattern or for a Brownian evolution threshold model[31]. Traits evolving under a Brownian model have $D = 0$. If the trait is highly conserved, the observed sum of changes along the phylogeny will be very low ($D < 0$). $D$ values between 0 and 1 are indicative of a trait less conserved than expected under Brownian motion. Phylogenetic randomness in trait evolution is shown by $D = 1$.

### Phylogenetic correlation

The phylogenetic correlation between male and female same-sex sexual behaviour was obtained using the approach from ref. [42] as implemented in the R package "diversitree"[63]. This correlation was run using the fitMK optimization procedure and the ARD model of evolution. To test for a significant correlation between both types of same-sex sexual behaviour we compared the likelihood of a model where male and female same-sex sexual behaviour are allowed to evolve

independently with that of a model where they evolve in a dependent way[42,64]. The model where male and female same-sex sexual behaviour are correlated was constructed by constraining the transition rates of one trait to be dependent of the state of the other trait, finally yielding four transition rates. The model where both types of same-sex sexual behaviour are uncorrelated has no constrains in any transition rate, and yields eight transition rates. The advantage of the method implemented in diversitree is that it can control for phylogenetic pseudor-eplication, the occurrence of significant association between two discrete traits even when the pattern is driven by a single (or, very few) independent transition(s) from one character state to another[65,66]. This analysis was performed by using the make.musse.multitrait function by making the argument depth as 0 for the uncorrelated model and depth = c(0,0,1) for the correlated model following the specifications from equation 5 in ref. 63. To account for phylogenetic uncertainty, this analysis was run using 30 randomly chosen trees.

### Ancestral reconstruction

Ancestral states reconstruction of male and female same-sex sexual behaviour was performed using maximum likelihood for discrete characters. We considered all possible transition rates between states to receive distinct parameters (ARD model of evolution). We obtained marginal probabilities for all nodes. To account for phylogenetic uncertainty, we reconstructed the state of all nodes using a set of 100 randomly selected phylogenies. All analyses were performed by means of the ace (Ancestral Character Estimation) function in the R package "ape"[67]. Statistical difference of each ancestral value from the theoretical value = 0.5 was performed by means of a z-score test.

We estimated the number of times that same-sex sexual behaviour have been independently gained and lost during the evolution of mammals using stochastic mapping[32]. Given an observed phylogenetic tree and distribution of character states, stochastic mapping generates multiple iterations of character evolution that are consistent with the observed character states, using a continuous time-reversible Markov model. We run 1000 simulations to infer the values of gains and losses. We performed this analysis using the make.simmap function in R package "phytools"[64].

We compared the average age of the internal nodes of the phylogeny where same-sex sexual behaviour was inferred as present with the average age of the nodes where it was inferred as absent. Because our mammalian phylogenetic tree is dated, we first assessed the age of each internal node. Afterward, we calculated for all the internal nodes the likelihoods of exhibiting same-sex sexual behaviour using the ace function in the R package "ape"[67]. We have repeated this procedure for 100 randomly chosen trees. To find out if each of these likelihoods unequivocally indicate whether the ancestral node exhibited same-sex sexual behaviour or not, we compared their values with the theoretical values obtained from a null model. Such a null model was built by randomly reshuffling 100 times the occurrence of same-sex sexual behaviour across the tips of the phylogeny and calculating for each internal node the likelihoods of exhibiting and not exhibiting same-sex sexual behaviour. The observed values above or below these theoretical values were considered to exhibit or not to exhibit same-sex sexual behaviour, respectively. To be conservative, we considered equivocal those nodes with values belonging to the interval between the theoretical values. This procedure was repeated for each of the 100 randomly chosen trees. We statistically compared then the age distribution of the nodes inferred as exhibiting same-sex sexual behaviour versus the age distribution of those inferred as not exhibiting same-sex sexual behaviour by means of a t-test.

### Testing adaptive hypotheses explaining the evolution of same-sex sexual behaviour

To test the predictions of each adaptive hypothesis we performed a series of phylogenetic models including presence of same-sex sexual

behaviour as dependent variables and adulticide and sociality as independent variables. We have used four complementary approaches to control for the potential bias caused by any effect caused by differences in research intensity:

**Method 1.** As a first way of controlling for research intensity, following ref. 35, we have included research effort, measured as the citation counts for each species[37,38], as a covariate in the phylogenetic models performed to test the hypotheses postulated to explain the evolution of same-sex sexual behaviour in mammals[37,38]. In this case we fitted phylogenetic logistic regressions[35] using as dependent variable the same-sex sexual behaviour as a binary trait (yes, no) and including adulticide (male adulticide when testing male same-sex sexual behaviour and female adulticide when testing female same-sex sexual behaviour), sociality and sampling effort as independent variables. These variables were weakly correlated ($\rho = 0.12 \pm 0.09$, mean $\pm 1$ standard deviation of pairwise Spearman's rank correlations), indicating that multicollinearity did not affect the interpretation of the analyses (variance inflation factor <2.8 in all cases). The parameters were obtained by 100 bootstraps, and the phylogenetic signal was simultaneously calculated after controlling for the independent variables, using alpha to estimate the level of phylogenetic correlation. We made separate models for males and females. These analyses were performed using the R package "phylolm"[68]. We applied this method to subsets I and II.

**Method 2.** As a second way of controlling for research intensity, following ref. 35, we have included research effort as a weighting factor in the phylogenetic models rather than as a covariate[40,41]. These models control for research intensity by giving to the scores of same-sex sexual behaviour of each species a statistical weight proportional to its citation counts. These weighted models were performed by means of Bayesian phylogenetic generalised linear models with binomial error distribution. This type of models allows weighting each sampling unit (each species) by its research effort while simultaneously calculating the phylogenetic signal for binary dependent variables (presence/absence of same-sex sexual behaviour). These analyses were performed using the R package "MCMCglmm"[36]. We applied this method to subsets I and II.

**Method 3.** As a third way of controlling for research intensity, we have performed the phylogenetic logistic regressions including only those species whose reproductive and sexual behaviour have been studied profusely (subset III). The fitted phylogenetic logistic regressions were fitted as explained in method I.

**Method 4.** As a fourth way of controlling for research intensity we performed the phylogenetic logistic regressions removing those species where its overall behaviour has been studied occasionally (subset IV). The fitted phylogenetic logistic regressions were fitted as explained in method I.

### Directional test of trait evolution

We tested whether the covariance of same-sex sexual behaviour with sociality and adulticide reflect dependency relationships between these behaviours or independent evolutionary processes in the same direction. For this, we used the Pagel' directional test of trait evolution[42]. For each sex (male and female) and each of the two pairs of behavioural traits (same-sex sexual behaviour vs. adulticidal behaviour and same-sex sexual behaviour vs. sociality), we compared four alternative models of evolution using the method described in ref. 42. We used in each model the adulticide committed by the same sex that was tested (when exploring same-sex sexual behaviour in males we included in the analyses male adulticide and when studying female same-sex sexual behaviour we included in the analyses female adulticide). These models were:

(1) A first model postulating that changes in the behavioural traits were independent of each other. These models were built up forcing transitions to be constrained to be independent.

(2) A second model that postulates that changes in the behavioural traits are interdependent.

(3) A third model that postulates that changes in same-sex sexual behaviour preceded changes in adulticide or sociality, suggesting that these two mammalian features did not cause same-sex sexual behaviour evolution. An ancestral species with no same-sex sexual behaviour and no adulticide/sociality evolved same-sex sexual behaviour, and as a consequence of this change in sexual behaviour, subsequently moved toward adulticide or sociality. Under these models the evolution of these two behaviours would depend on the same-sex sexual behaviour state. These models are built up by constraining the transitions in same-sex sexual behaviour on the other behavioural trait.

(4) A fourth model that postulates that changes in adulticide or sociality preceded changes in same-sex sexual behaviour, suggesting that those two characters drove evolution of same-sex sexual behaviour. An ancestral population moved first toward social or adulticidal state and afterwards evolved same-sex sexual behaviour. Under this hypothesis the evolution of same-sex sexual behaviour would depend on the state of adulticide/sociality. These models are built up by constraining the transitions in any of the two behaviours (adulticide and sociality) on same-sex sexual behaviours.

To assess which model can explain best the evolutionary pattern of same-sex sexual behaviour in mammals, we first performed likelihood ratio tests comparing the independent models against the other three models. These tests indicate if there is correlated evolution. Afterward, we decided which of the four models the empirical evidence best supports by comparing their Akaike weights[69]. Akaike weights can be directly interpreted as conditional probabilities for each model[70]. To control for phylogenetic uncertainty, we performed each of these analyses with 10 randomly chosen variations of the Faurby & Evenning phylogenetic tree[61]. All analyses were performed using the function fitPagel in R package phytools[64] with "fitDiscrete" as the optimization method and using the ARD model that allows all rates to differ.

### Reporting summary
Further information on research design is available in the Nature Portfolio Reporting Summary linked to this article.

## Data availability
The data used in this study are provided in the Supplementary Information and Supplementary Data 1 and 2. We downloaded information from:

 1) panTheria: https://ecologicaldata.org/wiki/pantheria.

 2) Amniote: https://datarepository.wolframcloud.com/resources/Amniote-Life-History-Database.

 3) EltonTraits 1.0: https://opentraits.org/datasets/elton-traits.html.

 4) Animal Diversity Web: https://animaldiversity.org/.

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

## Acknowledgements

Authors thank Drs. Juan Gabriel Martínez Suárez, Jesús Avilés and Eugene W. Schupp for helpful discussion on this topic. This paper has been funded by the project QUAL21-011 granted by the Consejería de Universidad, Investigación e Innovación of the Junta de Andalucía. This is a contribution to the UGR-CSIC Unidad Asociada "Evoflor".

## Author contributions

JMG designed the work, contributed to the acquisition, analysis, and interpretation of data, draughted the work, and approved the submitted version. AGM designed the work, contributed to the acquisition, analysis, and interpretation of data, draughted the work, and approved the

submitted version. MV designed the work, contributed to the acquisition, analysis, and interpretation of data, draughted the work, and approved the submitted version

## Competing interests

The authors declare no competing interests.
