## [Peer Review File · Nature Communications]

The evolution of same-sex sexual behaviour in mammalsEditorial Note: This manuscript has been previously reviewed at another journal that is not operating a transparent peer review scheme. This document only contains reviewer comments and rebuttal letters for versions considered at Nature Communications.

Reviewers' Comments:

Reviewer #2:

Remarks to the Author:

I think the authors have taken the comments on board and I would like to see this manuscript published. My only comment at this point is that as the revisions of this manuscript have got through the analyses have become a bit patchwork in my opinion. And I think that needs some attention.

For instance, the authors have shown that research intensity is important in their GLM type analyses- which is good. However, this is not considered in the Markov model type analyses. Thus, these maybe (or are very likely to be) bias by research intensity to some undetermined extent.

Ideally, I would suggest that the authors take a step-back and present an integrated set of analyses that deal with the issue of research intensity throughout. But at the very very least include a detailed and open discussion of the issue.

Reviewer #4:

Remarks to the Author:

I read the reviews, the response to the referees and the new version of the manuscript.

My main concern is about the scarcity of the dataset (S1) and the fact that inferences of dating, most common ancestor, etc could be due to the paucity of data. There are 321 species of mammals where same-sex sexual (SSB) behaviour was noted, most of it comes from anecdotal observations compiled in Bagemihl 1999. I just scanned through the list on Table S1, and for example saw that Barbary macaques are classified as "moderate SSB in semiwild conditions" where some of my students observed it to happen extensively in a semi wild population. A total of 321 species out of ca.4,000 mammals could be artefactual: absence of evidence is not evidence of absence, and, given that there have been no systematic studies of SSB in any group of mammals, I can't help but be very cautious about the results presented here.

I commend the author for trying, but I think data are simply not yet available for such a study. This was commented by referee 3 as [lack of] "research intensity". I see the authors are aware of this caveat and tried to address it by using a subset of the data (those species where SSB was observed in the wild and in captivity, and those species where SSB was observed in the wild), but that would, in my mind, only decrease the number of data points without correcting for the 'absence of evidence' bias. The very high number of missing data (rather than species where we know SSB is absent) could well explain why the authors found that closely related species do not necessarily share this sexual behaviour [SSB]. The authors agree with this problem and confirm "We are aware that this outcome may change if same-sex sexual behaviour is found in species in which, due to low sampling effort, have not yet been detected". Not surprisingly, for "the 22 mammalian species that are the subject of continuing long-term, individual based field studies, same-sex sexual behaviour has been found in more than 80%" - extending this figure to all mammalian species, or at least a much larger number than 321, and then all the results presented here may well change a lot.

Another area of concern is their presentation of the adaptive explanations for SSB. They present two main hypotheses, citing a rather old paper by Bailey & Zuk, that (i) SSB contributes to social cohesion; (ii) SSB contribute to diminishing conflicts. 'Adaptive' should really be taken in a more accurate sense of contributing to fitness benefits, either indirectly or directly. This is a well-established concept in SSB research, with theoretical and empirical examples in a range of species, which is missing here. One 'classical' example of an adaptive explanation for SSB is if males engaging in SSB look after the offspring of their sisters, thereby increasing their reproductive output. These explanations should be discussed even though I cannot see how they could be tested here. As for what was tested here, like one of the referees, I am not sure I understand the logic of SSB being more frequent in species with aggressive and lethal intrasexual interactions – if (ii) SSB contributes to diminishing conflicts, why would you not expect it to be common in species where conflicts are rare, in fact the Bonobos are a perfect example of this. I see the authors tried to test for this looking at what came first, aggression or SSB, but again I do not think there are enough reliable data to apply such tests.

I am sorry I cannot be more positive, as I certainly agree with the authors that SSB is not maladaptive nor aberrant. Maybe a detailed study in primates, where we have the largest amount of SSB data, but still following more extensive 'gap filling' research, would make more sense at this stage.

Reviewer #2

I think the authors have taken the comments on board and I would like to see this manuscript published. My only comment at this point is that as the revisions of this manuscript have got through the analyses have become a bit patchwork in my opinion. And I think that needs some attention.

This is unfortunate, and we have tried to avoid it during the entire process. However, it is not always possible to keep internal consistency in a manuscript when it passes through several rounds of reviews. Nevertheless, we have tried to integrate all the approaches we have used in this study to make the manuscript more coherent. Please see below our response to your recommendation.

For instance, the authors have shown that research intensity is important in there GLM type analyses- which is good. However, this is not considered in the Markov model type analyses. Thus, these maybe (or are very likely to be) bias by research intensity to some undetermined extant.

Ideally, I would suggest that the authors take a step-back and present and integrated set of analyses that deal with the issue of research intensity throughout. But at the very very least include a detailed and open discussion of the issue.

Following the recommendation of Reviewer #2, we have re-run all the phylogenetic analyses using the four subsets of species created to control for research effort and intensity. As Reviewer #2 can observe, the results are very consistent across all subsets, and the outcomes and conclusions have not changed. We thank Reviewer #2 for this recommendation, since it has improved the soundness and robustness of our findings without changing the message of the study. In addition, now the manuscript is more integrated because we have used the same method to control for research intensity throughout the manuscript.

Reviewer #4

My main concern is about the scarcity of the dataset (S1) and the fact that inferences of dating, most common ancestor, etc could be due to the paucity of data. There are 321 species of mammals where same-sex sexual (SSB) behaviour was noted, most of it comes from anecdotal observations compiled in Bagemihl 1999. I just scanned through the list on Table S1, and for example saw that Barbary macaques are classified as “moderate SSB in semiwild conditions” where some of my students observed it to happen extensively in a semi wild population. A total of 321 species out of ca.4,000 mammals could be artefactual: absence of evidence is not evidence of absence, and, given that there have been no systematic studies of SSB in any group of mammals, I can't help but be very cautious about the results presented here.

We agree with Reviewer #4 that this is an important caveat not only in this study but in any study using macroevolutionary data, since it is almost impossible to be sure that

the available information is complete. We have been discussing about this potential caveat during the four rounds of reviews that we have had on this manuscript. To overcome this problem, we have conducted our analyses by controlling the research effort and intensity. The Reviewer can check the supplementary information where he/she could appreciate our great effort in controlling for the potential biases caused by the paucity of data.

We have explicitly stated throughout the manuscript that findings could change as more data are collected, we want to call the attention of Reviewer #4 that we already indicated this in several places:

L204-205: We are aware that this outcome may change if same-sex sexual behaviour is found in species in which, due to low sampling effort, have not yet been detected.

L216-218: We expected some equivocal reconstruction as a consequence of the scarce information on same-sex sexual behaviour existing for many mammal species

L316-319: This figure is probably underestimating the actual prevalence of same-sex sexual behaviour in mammals, since this behaviour has attracted the attention of behavioural ecologists and evolutionary biologists only recently^{1-4,10} and it is probably underreported¹¹.

L322-325: All of this makes evident the necessity of increasing the number of studies on this overlooked behaviour and the convenience of controlling for several sources of uncertainty in order to reliably determine the prevalence of same-sex sexual behaviour in mammals.

L339-341: We fully recognize that these results may change in the future if same-sex sexual behaviour is studied more intensively and comes to be detected in many more species^{7, 25}. This again vindicates the need to study this sexual behaviour more profusely in mammals.

Despite of this, to ensure that we make very clear that our findings are dependent on the available information, we have added the following cautionary notes:

L358-360: Albeit acknowledging that these findings may change if more data on same-sex sexual behaviour are reported.

L426-427: We recognize that this hypothesis could be partially modified by gathering more information on same-sex sexual behaviour in mammals.

By adding these new sentences, we have included cautionary notes in five out of the eight paragraphs of our Discussion, including the conclusion paragraph. We hope we have made clear the limitation of our data.

We honestly believe that we have been very cautious. In fact, we have added cautionary notes in five out of the eight paragraphs of our Discussion, including the conclusion paragraph. We are convinced that readers will appreciate our caution in interpreting our results.

I commend the author for trying, but I think data are simply not yet available for such a study. This was commented by referee 3 as [lack of] “research intensity”. I see the authors are aware of this caveat and tried to address it by using subset of the data (those species where SSB was observed in the wild and in captivity, and those species where SSB was observed in the wild), but that would, in my mind, only decrease the number of data points without correcting for the ‘absence of evidence’ bias. The very high number of missing data (rather than species where we know SSB is absent) could well explain why the authors found that closely related species do not necessarily share this sexual behaviour [SSB]. The authors agree with this problem and confirm “We are aware that this outcome may change if same-sex sexual behaviour is found in species in which, due to low sampling effort, have not yet been detected”. Not surprisingly, for “the 22 mammalian species that are the subject of continuing long-term, individual based field studies, same-sex sexual behaviour has been found in more than 80%” – extending this figure to all mammalian species, or at least a much larger number than 321, and then all the results presented here may well change a lot.

We agree with Reviewer that in many cases data are not simply available. However, we disagree in that using subsets only *decrease the number of data points without correcting for the ‘absence of evidence’ bias*. When subsets are done using a standardized approach, it contributes to remove false negatives (the big problem indicated by the Reviewer in her/his review), making comparisons more robust. And if the different subsets give similar outcomes, the process of hypothesis testing is more accurate and researchers can be more confident that the observed evidence is strong. As Reviewer #4 can observe, we have repeated all the analyses (both the phylogenetic and the comparative analyses) using the same method of control for research intensity.

Nevertheless, as the Reviewer has noted, we have been always very cautious, explicitly stating that our results highly depend on the available information on SSB. We hope our study will contribute to increase the interest for studying SSB in mammal species other than primates, in order to get a more complete picture on how this behaviour could have evolved.

Another area of concern is their presentation of the adaptive explanations for SSB. They present two main hypotheses, citing a rather old paper by Bailey & Zuk, that (i) SSB contributes to social cohesion; (ii) SSB contribute to diminishing conflicts. ‘Adaptive’ should really be taken in a more accurate sense of contributing to fitness benefits, either indirectly or directly. This is a well-established concept in SSB research, with theoretical and empirical examples in a range of species, which is missing here. One ‘classical’ example of an adaptive explanation for SSB is if males engaging in SSB look after the offspring of their sisters, thereby increasing their reproductive output.

These explanations should be discussed even though I cannot see how they could be tested here. As for what was tested here, like one of the referees, I am not sure I understand the logic of SSB being more frequent in species with aggressive and lethal intrasexual interactions – if (ii) SSB contributes to diminishing conflicts, why would you not expect it to be common in species where conflicts are rare, in fact the Bonobos are a perfect example of this. I see the authors tried to test for this looking at what came first, aggression or SSB, but again I do not think there are enough reliable data to apply such tests.

Reviewer is right when indicating that there are more adaptive explanations (understanding adaptive at macroevolutionary level as any trait -including behaviour traits- that has been evolved by natural selection to perform the function that it is currently performing). In macroevolutionary studies, in contrast to microevolutionary studies, good evidence of the adaptive nature of a given trait happens when there is an association between the presence of the trait and the performance of a given function or the occurrence of another trait that is postulated as the selective factor triggering the evolution of the target trait. Of course, the only hypotheses that can be tested are those that include information for the studied set of species. It would be nice to have information on other traits and functions, like the one suggested by the Reviewer, for many species. Unfortunately, as the Reviewer indicates, we do not have that information and consequently the hypothesis, although potential true, cannot be tested. Nevertheless, we explicitly say in the conclusion paragraph that our findings do not preclude the contribution of other proximate mechanisms, causing the display of same-sex sexual behaviour in some specific species. Again, we are very cautious when interpreting our results.

I am sorry I cannot be more positive, as I certainly agree with the authors that SSB is not maladaptive nor aberrant. Maybe a detailed study in primates, where we have the largest amount of SSB data, but still following more extensive 'gap filling' research, would make more sense at this stage.

I appreciate this proposal of studying primates, However, we want to remark that one of the main strengths of our study is having explored the evolution of SSB at the level of the entire Mammalia class.